# *Bacillus thuringiensis* Bioinsecticides Induce Developmental Defects in Non-Target *Drosophila*
*melanogaster* Larvae

**DOI:** 10.3390/insects11100697

**Published:** 2020-10-13

**Authors:** Marie-Paule Nawrot-Esposito, Aurélie Babin, Matthieu Pasco, Marylène Poirié, Jean-Luc Gatti, Armel Gallet

**Affiliations:** Université Côte d’Azur, CNRS, INRAE, ISA, UMR CNRS 7254/INRAE 1355/UCA, 400 route des Chappes, BP 167, 06903 Sophia Antipolis CEDEX, France; marie-paule.esposito@inrae.fr (M.-P.N.-E.); aurelie.babin@hotmail.fr (A.B.); armel.gallet@inrae.fr (M.P.); Marylene.Poirie@unice.fr (M.P.); jean-luc.gatti@inrae.fr (J.-L.G.)

**Keywords:** *Drosophila melanogaster*, *Bacillus thuringiensis*, developmental defects, gut cell homeostasis, *Lactobacillus plantarum*, bioinsecticides

## Abstract

**Simple Summary:**

The use of chemical pesticides is controversial mainly because of their detrimental effects on both the environment and human health. Public opinion along with government policies encourage reductions in their exploitation. An emerging solution is the use of biopesticides, which sales are currently increasing. Among biopesticides, the bacterium called *Bacillus thuringiensis* is the most used bioinsecticide both in organic and conventional farming to fight larval pests. One strain of *Bacillus thuringiensis* produces toxins that specifically kill lepidopteran (butterfly) larvae in two or three days. However, although not lethal for other insects such as bees or fruit flies, *Bacillus thuringiensis* may affect their development. Only scarce data are currently available on the unintended effects of *Bacillus thuringiensis* strains on insects other than those targeted. In this study, we characterized the adverse effects of *Bacillus thuringiensis* strains that target lepidopteran larvae, on the development of the fruit fly *Drosophila*
*melanogaster*, a dipteran non-target insect. We showed that amounts of *Bacillus thuringiensis* in the nutritive medium close to the amounts found on vegetables after treatment induced developmental and growth defects in *Drosophila* larvae. We further showed that these effects are due to the disturbance of the larval intestinal physiology, reducing protein digestion.

**Abstract:**

Bioinsecticides made from the bacterium *Bacillus thuringiensis* (*Bt*) are the bestselling bioinsecticide worldwide. Among *Bt* bioinsecticides, those based on the strain *Bt* subsp. *kurstaki* (*Btk*) are widely used in farming to specifically control pest lepidopteran larvae. Although there is much evidence of the lack of acute lethality of *Btk* products for non-target animals, only scarce data are available on their potential non-lethal developmental adverse effects. Using a concentration that could be reached in the field upon sprayings, we show that *Btk* products impair growth and developmental time of the non-target dipteran *Drosophila melanogaster*. We demonstrate that these effects are mediated by the synergy between *Btk* bacteria and *Btk* insecticidal toxins. We further show that *Btk* bioinsecticides trigger intestinal cell death and alter protein digestion without modifying the food intake and feeding behavior of the larvae. Interestingly, these harmful effects can be mitigated by a protein-rich diet or by adding the probiotic bacterium *Lactobacillus plantarum* into the food. Finally, we unravel two new cellular mechanisms allowing the larval midgut to maintain its integrity upon *Btk* aggression: First the flattening of surviving enterocytes and second, the generation of new immature cells arising from the adult midgut precursor cells. Together, these mechanisms participate to quickly fill in the holes left by the dying enterocytes.

## 1. Introduction

The use of conventional synthetic chemical pesticides is controversial mainly because of their harmful effects on human health and ecosystems. Public opinion along with governmental policies encourage reduction in their use [1,2]. An emerging solution is their replacement with biopesticides.

Among bioinsecticides, *Bacillus thuringiensis* (*Bt*) products are increasingly sprayed in both organic farming and conventional agriculture [3]. *Bt* products are the second most used insecticides in the world with 32,000 tons sold in 2017 [2]. *Bt* is a gram-positive spore-forming bacterium belonging to the *Bacillus cereus* group [4]. It was first identified and characterized for its specific entomopathogenic properties due to the presence of Cry toxins, which are produced in a crystalline form during the sporulation of bacteria [5]. Seventy-eight different strains of *Bt* are currently inventoried [6], producing a total of more than 300 distinct Cry toxins [7] with a spectrum of toxicity ranging from nematodes to human tumor cells [8,9]. However, only four are used commercially as bioinsecticides owing to the specific acute toxicity of their Cry toxins to pest larvae: *Bt* subsp. *kurstaki (Btk)* and *Bt* subsp. *aizawai* to kill lepidopteran larvae, *Bt* subsp. *israelensis* to kill mosquito larvae and *Bt* subsp. *morrisoni* to kill coleopteran larvae [10]. Upon ingestion of spores and crystals, Cry protoxins are first released from the crystal before being processed by gut proteases to give rise to the active Cry toxins. Then, these Cry toxins bind to receptors at the surface of the intestinal cells, inducing their death and ultimately the destruction of the gut epithelium. The spores germinate and, thanks to the holes made by Cry toxins in the gut lining, vegetative bacteria invade the internal cavity of the body, causing sepsis that kills the targeted pest in 2 or 3 days. The specificity of acute Cry toxicity relies on both the capacity of gut proteases to cleave and activate Cry protoxins and on the presence of host receptors specifically recognized by Cry toxins [8]. Each subspecies of *Bt* produces between one and six different Cry toxins. In general, all the Cry toxins contained in a *Bt* subspecies target a specific phylogenetic order (e.g., lepidopteran, coleopteran, dipteran) [11]. The most used *Bt* subspecies in forestry and agriculture to fight lepidopteran larvae is *Btk*, which produces five different Cry toxins (Cry1Aa, Cry1Ab, Cry1Ac, Cry2Aa and Cry2Ab) [12].

The increasing environmental dispersion of *Btk* bioinsecticides and the persistence of the spores in the environment [13,14,15,16] raise the question of their long-term putative effects on both health and environment. There are many data sets showing that *Btk* spores and Cry toxins do not display harmful effects against non-target organisms [17,18]. Nevertheless, there are also studies suggesting that *Btk* spores and crystals may have unintended impacts on non-target animals. Indeed, studies have highlighted that Cry toxins can delay the development and increase the mortality of trichopteran species whose larvae have an aquatic lifestyle [19,20]. *Btk* bioinsecticides are also acutely toxic against *Trichogramma chilonis* females, a tiny egg endoparasitoid hymenopteran used as a biocontrol agent in integrated pest management to control lepidopteran pests on crops [21] and delay the imago emergence of many *Drosophila* species [22]. In vertebrates, Grisolia and colleagues observed that the exposure of *Danio rerio* fry to different *Btk* Cry toxins induced lethality and developmental delay [23]. Finally, many bacteria belonging to the *B. cereus* group are opportunistic or potent pathogens displaying virulence against a wide range of organisms [24].

Based on these previous studies, we decided to use the model organism *Drosophila melanogaster*, not typically targeted by *Btk*, to identify and characterize the putative impacts of *Btk* spores and toxins on development. While the acute toxicity of the agricultural doses of *Btk* bioinsecticides efficiently kills lepidopteran larvae, *Drosophila* larvae, being tolerant to these doses, allow the study of non-lethal impacts. In addition, *D. melanogaster* has been used successfully in hundreds of studies to decipher host-pathogen interactions and is a toolbox widely used to identify disturbed cellular and molecular mechanisms [25,26,27].

Here, using two realistic concentrations, a concentration detected on vegetables after one spraying and a concentration only tenfold higher (but equivalent to the ten cumulative treatments authorized by the European Union), we observed that *Btk* bioinsecticides induced developmental delay and reduced growth of *D. melanogaster* larvae. We further showed that *Btk* bioinsecticides triggered intestinal cell death and altered protein digestion without modifying the food intake and feeding behavior of the larvae. Interestingly, we showed that a protein-rich diet or supplementing food with the probiotic *Lactobacillus plantarum* (*L. plantarum*), known to enhance protein uptake [28], mitigated the impacts of *Btk* bioinsecticide on growth and development. We also found that *Btk* products did not interfere with the commensal flora of the larvae. Finally, we unraveled new cellular mechanisms for maintaining intestinal integrity mounted by the larval midgut epithelium in response to *Btk* aggression*,* particularly the flattening of enterocytes and the production of new cells from the adult midgut precursors. Overall, our data show that although not lethal for larvae of the non-target organism *Drosophila melanogaster*, agricultural doses of *Btk* bioinsecticides damage gut epithelium and consequently impair larval growth and development time.

## 2. Materials and Methods

### 2.1. Fly Stocks and Genetics

Canton S (WT) (Bloomington #64349); *w, dlg::GFP cc01936* (FlyTrap) was provided by A. Spradling; *w; myo1A-Gal4* was a gift from N. Tapon; *w, UAS-GC3Ai^G7S^* were kindly provided by Magali Suzanne and described in [29]. Apoptosis was monitored on F1 progeny *w/w; myo1A-Gal4/+; UAS-GC3Ai^G7S^/+* L3. The ReDDM fly stock to trace cell progenies in the midgut was kindly provided by Maria Dominguez [30]. To perform lineage tracing, synchronized eggs (*w/+; esg-Gal4 UAS-GFP/+; tub-Gal80^ts^ UAS-H2B::RFP/+*) were hatched on contaminated food (see below for the method) and maintained for 24 h at 25 °C. Then, L1 larvae were transferred and maintained at 29 °C to alleviate the inhibitory effect of Gal80^ts^ on the Gal4 transcription factor. Intestines were dissected and observed at mid L3 (between days 4 and 5).

### 2.2. Bacterial Strains and Commercial Bioinsecticides

*Lactobacillus plantarum^WJL^* that has been described in [31] and sequenced in [32] was kindly provided by Bernard Charroux (IDBM, Marseille, France) and by Renata Matos (F. Leulier’s Lab, IGFL, Lyon, France). *Pseudomonas entomophila* [33] was kindly provided by Bruno Lemaitre’s Lab (EPFL, Lausanne, Switzerland). *P entomophila* and *L. plantarum^WJL^* were grown as described in [34]. The *Btk^ΔCry^* strain (identified under the code 4D22) [35] was obtained from the *Bacillus* Genetic Stock Center (www.bgsc.org). Delfin and Dipel formulated bioinsecticides were bought in an agricultural shop center. The *Btk* strains (serotype 3a3b) used in Delfin and Dipel are called SA-11 and ABTS-351, respectively. Importantly, we noticed that a given commercial brand sold by retailers can contain either Dipel or Delfin. The strain *Btk* ABTS-351 was isolated from Dipel on LB (Lysogeny Broth) agar. Below are presented the different links to upload the Technical and Safety Data Sheets provided by the manufacturers for Delfin and Dipel [36,37,38]. Briefly, Delfin contains 85% of *Btk* spores (4.85 × 10^10^ UFC/g) and toxins + 15% of naphthalene sulfonic acid and sodium salt. Dipel contains 54% of *Btk* spores (1.17 × 10^10^ UFC/g) and toxins + 46% of “Other Trade Secret ingredients” (comprising probably 14% of sodium sulfate).

### 2.3. Spore Preparation

*Btk* vegetative cells were left to grow and sporulate for two weeks at 30 °C with shaking in 2 L of PGSM medium. The culture was then heated for 1 h at 70 °C to eliminate vegetative cells and centrifuged for 15 min at 5000× *g*. The pellet was first washed with 1 L of 0.15 M NaCl and twice with sterile water. The final pellet was suspended in 30 mL of sterile water, aliquoted and lyophilized for 24–48 h. The spore titration was determined by serial dilution of a known weight of lyophilized spores and counting the number of “colony forming units” (CFU) on LB agar after incubation at 30 °C for 18 h.

### 2.4. Crystal Purification

Two grams of the commercial product were suspended in 60 mL of sterile water, dissolved for 5 h at 4 °C with agitation, and then sonicated four times for 15 s with a frequency set at 50% (Fisherbrand™ Model 505 Sonic Dismembrator, Waltham, MA, USA). Ten milliliters of the homogenate were deposited on a step density gradient of 67/72/79% sucrose and centrifuged at 100,000× *g* for 18 h at 4 °C. Crystals were collected on the interfaces between the different sucrose phases and washed twice with sterile water. The final pellet was suspended in 12 mL of sterile water, aliquoted and lyophilized for 24–48 h.

### 2.5. Diet Composition

*Drosophila* were reared at 25 °C on conventional medium (referred as the “protein-poor” medium in this study): 8% cornmeal (AB Celnat), 2% yeast (Springaline^®^ BA10/0-PW, Amcan, Dusseldorf, Germany), 2.5% sucrose, 0.8% agar (ref 20768-361, VWR) and 0.6% methylparaben (Tegosept, ref 789063 Dutscher). The protein-rich medium contained 10% yeast instead of 2%.

### 2.6. Intestinal CFU Counting and Commensal Flora Estimation

Colony forming unit counting were described in [34]. Ten flies or ten mid-third instar larvae were washed first with 70% ethanol for 30 s, then washed with PBS and crushed in 500 µL of LB medium with a Tissue Lyser (Qiagen, Hilden, Germany, ref. 85600) at 50 Hz for 5 min. The homogenate was plated on different selective media to identify the common commensal bacteria. MRS agar (Sigma, St. Louis, MO, USA, ref 69964 + 1 mL of Tween 20/L) at 37 °C in anaerobic conditions was used to identify *Lactobacillus plantarum* and *L. brevis*. BHI agar (Sigma ref 70138) at 37 °C allowed the identification of *Enterococcus faecalis*. LB agar (Fisher, Hampton, HN, USA, ref BP1426-2) at 37 °C allowed the identification of *Escherichia coli* and *Staphylococcus aureus*. Mannitol medium (Bacto peptone 3 g/L, Yeast extract 5 g/L, D-Mannitol 25 g/L, agar 15 g/L) at 30 °C allowed the identification of *Commensalibacter intestini* and *Acetobacter pomorum*.

### 2.7. Axenic Flies

Adult Canton S flies were left to oviposit eggs for two days at 25 °C and 12/12 h day/night cycle on our conventional medium supplemented with a cocktail of four antibiotics: Ampicillin, kanamycin, tetracycline, and erythromycin (50 µg/mL each). In the next generation, newly hatched adults were immediately removed and transferred to fresh conventional medium containing the four antibiotics. This operation was repeated until flies with axenic gut were obtained. The presence of commensal flora was checked at each fly generation (Appendix A). Axenic flies were obtained after six generations.

### 2.8. Oral Infection of Larvae

Adult flies were left to oviposit eggs for 4 h at 25 °C on laying nests containing grape juice medium (25% grape juice, 3% agar, 1.2% sucrose, 2% ethanol, 1% acetic acid). Twenty eggs were collected and transferred to a *Drosophila* vial containing 2 g of the desired medium mixed with *Btk* spores, *Btk* crystals or the naphthalene sulfonic acid additive. The conditions used for 2 g of medium and 20 eggs were: 10^7^ CFU of *Btk* spores for the 1X concentration; 10^8^ CFU of *Btk* spores for the 10X concentration; 0.67 mg of *Btk* purified crystals (equivalent to the 10X concentration) and considering that crystals represents 15–30% of spore weight [39,40,41]; 10^8^ CFU of *L. plantarum* or *P. entomophila*; 0.4 mg of naphtalene-2-sulfonic acid sodium salt (ref A13066, Alfa Aesar, Haverhill, MA, USA) (equivalent to the 10X concentration). Vials were incubated at 25 °C with a 12/12 h day/night cycle until further processing.

### 2.9. Pupation Curve

Immobilized larvae forming white pupae were counted. The number of pupae was expressed as the percentage of the final number of pupae. While 20 eggs were deposited in each tube, in some conditions, especially with Dipel and Delfin, we never got 20 pupae probably due to previously described larval lethality induced by the *Btk* products [22]. Table 1 summarizes the developmental time at which 10%, 50% and 90% of the larvae were immobilized for all the conditions we tested. For each condition described above, the experiment was performed at least in three experimental blocks of triplicates.

### 2.10. Pupal Size

On day 7, 8 and 9 after egg laying, pupae were gently removed from the vial, washed in water and slightly dried with tissue paper. Pupae were placed on a microscope slide and pictures were taken with a numeric microscope VHX-2000 (Keyence, Osaka, Japan). Pupal length and width were measured with Image J software and pupal volume was calculated using the following formula for ellipsoid: V = [(4/3) × 3.14 × (L/2) × (l/2)^2^] where V is the volume, L the length and l the width of the pupa.

### 2.11. Larval Feeding Behavior and Food Intake

Quantification of larval food intake was performed according to [42] with the following modifications: We deposited a mix of 50% yeast/0.75% of the food blue dye Erioglaucin (ref 861146, Sigma) in the center of a Petri dish lid (diameter: 35 mm) to feed 20 mid-third larvae (previously infected or not) *ad libitum*. After 2 h at 25 °C, larvae were collected and washed with water. The number of white larvae (indicating no feeding) and blue larvae was counted. Ten blue larvae were transferred into a microtube containing 500 µL of PBS, crushed at 50 Hz for 2 min with a TissueLyser (Qiagen, ref 85600) and centrifuged at 10,000× *g* for 5 min at 4 °C. The amount of Erioglaucin in the supernatant was quantified at 629 nm with a spectrophotometer (SpectraMax^®^ Plus384, Molecular device, San Jose, CA, USA).

### 2.12. Midgut Permeability

The same technique as described above was used except that the mix contained 50% yeast and 1.25 mg/mL Dextran^FITC^ (ref 46944, Sigma) in water. Twenty previously orally infected mid-third instar larvae were added. After 2 h at 25 °C, the larvae were collected, washed with water and placed for at least 30 min at 4 °C. Pictures were taken with a fluorescent stereoscopic AxioZoom V16 (Zeiss, Oberkochen, Germany) microscope.

### 2.13. Metabolic Assay

Intestinal protease activity assay was performed as described in [43] with eight mid-third instar larval midguts per condition. Lipids were assayed as described in [44]. The tri- and diacylglycerol (TAG and DAG) were measured on 3 µL of hemolymph obtained from 5 day old larvae using Triglyceride Reagent (ref T2449, Sigma) and Free Glycerol Reagent (ref F6428, Sigma). The metabolite quantity was reported per mg of protein or per µL of hemolymph. Protein assay was achieved with Biorad Protein Assay Dye Reagent Concentrate (ref 5000006, Biorad, Hercules, CA, USA). All experiments were performed at least three times in triplicate.

### 2.14. Lipid Labeling

Five days after infection (mid-third instar larvae), ten larval guts were dissected in PBS and fixed for 40 min in 4% formaldehyde in PBS. Guts were then washed twice in PBS and further incubated for 30 min either in 5 µg/mL of BODIPY™ 493/503 (ref D3922, Invitrogen^TM^ Molecular Probes^TM^, Waltham, MA, USA) or in a 0.06% Oil-red-O solution (ORO) (ref 189400250, Acros Organics, Fair Lawn, NJ, USA). Guts were then washed twice with PBS. For BODIPY™ labeling guts were mounted in Fluoroshield/DAPI (ref F6057, Sigma) and immediately observed under a fluorescent microscope (Zeiss Axioplan Z1 with Apotome 2). For Oil-red-O labeling, guts were mounted in 80% glycerol/PBS and observed with a numeric microscope (VHX 2000, Keyence).

### 2.15. Immunohistochemistry, Image Capture and Processing

Ten guts of late third instar larvae were dissected in PBS and fixed for 40 min in 4% formaldehyde in PBS. Guts were then washed twice with PBS-0.1% Triton X100, incubated for 3 h in Blotto (2.5% FBS, 0.1% Triton X-100, 0.02% sodium azide in PBS) and further incubated with the primary antibody overnight at 4 °C in Blotto. The following dilutions were used: Mouse anti-Armadillo (DSHB #N2 7A1) at 1/50 and Rabbit anti-Phospho-histone-3 (ref 9701S, Cell signaling, Danvers, MA, USA) at 1/500. Guts were washed once with Blotto before incubation in the secondary antibody and/or phalloidin (2 h at room temperature in Blotto). The secondary antibodies were the following: AlexaFluor^TM^ 546 donkey anti-Rabbit IgG (ref A10040, Invitrogen) used at 1:500; AlexaFluor^TM^ 647 goat anti-Mouse IgG (H + L) (ref A21235, Invitrogen) used at 1/500 and Alexa Fluor™ 555 Phalloidin (ref A34055, Invitrogen) used at 1/1000. Guts were washed several times in PBS, mounted in Fluoroshield/DAPI (ref F6057, Sigma) and observed with a Zeiss Axioplan Z1 with Apotome 2 microscope. Images were analyzed using ZEN (Zeiss) software or Photoshop. For cell death monitoring (in *myo1A-Gal4 UAS-UAS-GC3Ai^G7S^* larvae), guts were dissected in PBS, mounted in 80% Schneider medium/20% glycerol and pictures were immediately taken with a fluorescent stereoscopic microscope (AxioZoom V16, Zeiss). Image acquisition was performed at the microscopy platform of the Institut Sophia Agrobiotech (INRAE 1355-UCA-CNRS 7254-Sophia Antipolis).

### 2.16. Polyploidy Estimation

Ploidy was calculated by measuring the area of the nuclei of late L3. Only round nuclei were analyzed. The average nucleus area of diploid adult midgut precursor cells was calculated. This average area (8 µm^2^) was assigned to the 2C ploidy (C corresponding to chromatin copy number). The theoretical average area of 4C, 8C, 16C, 32C and 64C nuclei was calculated on the basis of the measured 2C average area. Nuclei were counted (109 for the control larvae, 184 nuclei for *Btk^ΔCry^* 10X larvae and 306 nuclei for Dipel 10X larvae) and binned into ploidy groups.

### 2.17. Statistics

When “*n*” was equal or superior to 30, statistical analysis was performed using a parametric *t*-test. An *F*-test was systematically done before applying the *t*-test to verify the homogeneity of variances. When “*n*” was inferior to 30, we used the non-parametric pairwise comparisons of the Wilcoxon-Mann-Whitney test. Significance symbols are the following: *** (*p* ≤ 0.001); ** (*p* ≤ 0.01), * (*p* ≤ 0.05).

## 3. Results

### 3.1. Btk Bioinsecticide Induces Developmental Delay and Growth Defects

European and French Food Safety Agencies (Efsa and Anses, respectively) have estimated that the amount of *Btk* spores on fruit and vegetables after one treatment can reach 5 × 10^6^ Colony Forming Units (CFU)/g [45,46]. This quantification may be underestimated since weekly sprays are recommended by the manufacturer and up to 10 are authorized [46] (see also the European directives SANCO/1541/08—rev. 4 and SANCO/1543/08—rev. 4), knowing that the half-life of the entomopathogenic Cry1 toxins is estimated at ten days in the field [47]. Therefore, we decided to assess the impacts of the ingestion of two *Btk* bioinsecticides (i.e., Delfin and Dipel) on the growth and development of *Drosophila* larvae at two field-realistic concentrations: 5 × 10^6^ CFU/g (corresponding to the concentration observed after one spraying) and 5 × 10^7^ CFU/g (a concentration which could be reached upon repeated treatments), called hereafter “1X” and “10X”, respectively. We first measured the timing of larval development by performing a pupation curve. In the control experiment (H_2_O), 90% of the larvae pupated at 169 h after egg laying (this threshold will serve as a point of comparison throughout our study, Table 1 summarizes all the data at 10%, 50% and 90% of pupation entry). The 1X and 10X concentrations of Delfin postponed pupation by 10.5 h (+6.2%) and 16 h (+9.5%), respectively (Figure 1A and Table 1). We then assessed the impact of the second bioinsecticide Dipel. Because developmental delay was marked with Delfin at the 10X concentration, we used this same concentration as a benchmark for Dipel. Strikingly, the pupation delay increased to reach 25 h (+14.8%). Commercial products are made of spores, toxin crystals and additives (see Section 2). Dipel contains 46% “trade secret” components, while Delfin contains 15% of naphthalene sulfonic acid. To assess the potential involvement of the additives present in Dipel in the augmented delay observed, we plated the commercial product on LB agar, isolated a colony and prepared our own mixture of spores and toxins (the *Btk* strain used in Dipel is ABTS-351). When larvae were fed with the 10X concentration of the homemade preparation, we obtained developmental delays similar to those with the Delfin treatments (Figure 1A and Table 1), suggesting that the unknown additives present in the commercial preparation of Dipel might contribute to the increased delay we observed. Moreover, these data also suggested that the 15% naphthalene sulfonic acid present in Delfin had no impact on the developmental timing. Indeed, feeding developing larvae with an amount of naphthalene sulfonic acid equivalent to the 10X concentration of Delfin had no effect on the developmental timing (Figure 1B and Table 1). Then, we wondered whether the developmental delay was due to the spores alone, the crystals of toxins alone, or both. First, we fed the developing larvae with the 1X and 10X concentrations of spores of a *Btk* strain devoid of Cry toxins (*Btk^ΔCry^*). Whatever the concentration, no developmental delay was observed (Figure 1B and Table 1). Then, we fed the developing larvae with an equivalent 10X concentration of toxin crystals, considering that crystals represent 30% of the dry weight of bioinsecticides [39,40,41] (Appendix A). Again, we did not observe any developmental delay (Figure 1B and Table 1). Therefore, we assumed that the developmental delay we observed was mainly due to the mixture of spores and crystals (Figure 1A).

Since growth defects have a well-documented impact on developmental timing [48], we next measured the pupal volume, a known variable to estimate growth defects [49]. Interestingly, we observed a marked growth defect when the larvae were fed with Delfin and Dipel, as well as a weaker one when the larvae were fed with ABTS-351 (Figure 1C). We did not detect any growth defects in the other conditions. Together, our data show that ingestion of food contaminated with a mixture of *Btk* spores and crystals induced developmental delay and growth defects on the model organism *D. melanogaster*, at concentrations that can be found in treated fields.

Because we are interested in the host-pathogen interaction and that (i) the composition of Delfin is clearly provided by the manufacturer compared to Dipel (see Section 2), (ii) Delfin is composed of 85% of spores and toxin crystals while only 54% for Dipel, and (iii) naphthalene sulfonic acid (the remaining 15% in the Delfin formulation) has no effect on the developmental timing and growth, we decided to use Delfin in the rest of the study.

### 3.2. Btk Bioinsecticide Does Not Alter Feeding Behavior and Food Intake

The *Btk* load in the midgut could explain the differences observed between *Btk^ΔCry^* and Delfin. When we estimated the amount of *Btk* cells in the midgut of third instar larvae (L3) (Figure 2A), no difference was observed for 1X or 10X concentrations of Delfin compared to the equivalent concentrations of *Btk^ΔCry^*. Therefore, the variation in the quantity of *Btk* bacterial cells in the midgut could not explain the disturbed developmental timing and growth observed with Delfin.

Adult flies and larvae can sense the presence of harmful microbes, displaying an aversion to contaminated food [50,51,52,53]. Therefore, we tested whether feeding behavior was affected by the presence of Delfin. We added a blue dye in the food and counted the number of larvae containing blue food in their gut, either in L3 (Figure 2B) or in L2 stage (Figure 2C). We did not detect any change in feeding behavior for both L2 and L3 except a slight decrease for L2 larvae reared under Delfin 1X condition. We then quantified the food intake and we did not detect any significant modification in the L3 (Figure 2D), unlike in the L2, in which food intake was reduced whatever the condition (except for the *Btk^ΔCry^* 10X condition) compared to the control (Figure 2E). Therefore, although food contaminated with *Btk* spores (with or without crystals of toxins) disturbed food intake of L2 but not of L3, these data do not explain the specific impact of Delfin on the developmental time and growth of the larvae.

### 3.3. Btk Bioinsecticide Impairs Intestinal Protein Metabolism

Proteins are a key nutrient to drive normal larval development and growth. Increasing protein content in the diet accelerates the developmental time and/or stimulates growth, while an excess of sugar delays development and a change in lipid amounts has only a marginal impact [54,55,56,57,58,59]. Because we have previously shown that the presence of opportunistic bacteria in the adult midgut impede the digestion of proteins [34], we wondered whether this could also be the case in larvae in the presence of *Btk* bioinsecticide. While the 10X concentration of *Btk^ΔCry^* had no consequence, we observed a decreased capacity of the midgut of L3 to digest proteins upon Delfin 10X condition (Figure 3A).

We next wondered whether increasing the protein content of the diet could rescue the developmental delay and the growth defects caused by the ingestion of Delfin. Our rearing medium contains 2% yeast as a protein source (see Section 2). Therefore, we performed a pupation curve with larvae raised on a medium containing 10% yeast with the 1X or 10X concentrations of Delfin (Figure 3B). Compared to the control, no difference in the developmental time was observed for the 1X concentration of Delfin and only a slight delay for the 10X concentration (Table 1). Noteworthy, the developmental time for all conditions on the protein-rich medium was shortened compared to the control on our conventional medium (light green curve in Figure 3B and Table 1). Similarly, the pupal volume (Figure 3C) was completely restored for the 1X concentration of Delfin and partially rescued for the 10X concentration of Delfin (5.9% volume loss instead of 11.9% in the poor-protein diet). Taken together, our data suggest that Delfin impairs protein digestion, and thus gut functions, which can be rescued by adding extra dietary protein.

### 3.4. The Commensal Bacterium L. plantarum Helps to Overcome Btk Bioinsecticide Impacts

It has been recently shown that the commensal bacterium *L. plantarum* promotes the juvenile systemic growth of *Drosophila* larvae and infant mice by stimulating digestion and absorption of proteins, and protects young individuals against sepsis upon undernutrition conditions [28,60,61,62,63]. Therefore, we wondered whether complementing our conventional medium with *L. plantarum* would compensate for the impacts of the 10X concentration of Delfin. First, we observed that adding *L. plantarum* in the conventional medium shortened the developmental time of control larvae (Figure 4A, purple curve compared to green curve and Table 1). Interestingly, the presence of *L. plantarum* fully counteracted the Delfin-dependent developmental delay (Figure 4, pink curve compared to red curve and Table 1). We next observed that the presence of *L. plantarum* during larval development in the conventional medium reduced the pupal size of the control larvae by 4.4% compared to the control without *L. plantarum* (Figure 4B, purple bar compared to green bar) but fully rescued the growth defects induced by the 10X concentration of Delfin (Figure 4B pink bar). Since ingestion of *L. plantarum* and its establishment in the midgut [34] might compete for the presence of *Btk* and not act on protein intake as expected, we also controlled the number of *Btk* CFUs in the absence or presence of *L. plantarum* in the midgut of L3 and we found no difference (Figure 4C). Taken together, our data demonstrated that stimulation of digestion and/or absorption of proteins by the commensal bacterium *L. plantarum* helped to overcome the adverse effects of Delfin bioinsecticides on larval development.

### 3.5. The Impacts of Btk Bioinsecticide Do Not Rely on Commensal Bacteria Disturbance

*L. plantarum* is a widespread commensal bacterium of the *Drosophila* midgut [64,65,66]. Because complementing the larval diet with *L. plantarum* rescued the developmental delay and growth defects induced by *Btk*, we wondered whether the indigenous commensal flora of larvae reared on our conventional medium (i.e., poor in protein) was inhabited by *L. plantarum*. Only a few *L. plantarum* and/or *L. brevis* (belonging to the family of *Lactobacillaceae*) were present in the commensal flora of our larvae (Figure 5A). The dominant bacteria family we identified belonged to *Acetobacteraceae* known to be common in flies feeding on sugary, acidic, and alcoholic food [64,67]. This observation was in agreement with the composition of our conventional rearing medium poor in protein but quite rich in simple sugar (i.e., sucrose, see Section 2) and it suggested that the *Btk* effects we observed were independent of a putative deleterious impact on the indigenous *L. plantarum* of our larvae.

However, we could not rule out the impact of the *Btk* bioinsecticide on other members of the commensal flora. To test this hypothesis, we generated an axenic fly strain (see Section 2). Axenic larvae (Figure 5A) were further fed on our conventional medium treated or not with the 10X concentration of Delfin. The developmental time of the control axenic larvae was slightly shorter than those of the conventionally reared (CR) larvae (i.e., non-axenic) (Figure 5B and Table 1). Nevertheless, axenic larvae infected with Delfin 10X displayed a 19 h developmental delay compared to the control axenic larvae (+11.7%) (Figure 5B and Table 1), which was similar to the delay observed with *Btk*-treated CR larvae (see Figure 1). Then, while axenic larvae gave rise to pupae with a similar volume to that of CR larvae (Figure 5C), Delfin 10X still induced a reduction of pupal volume in axenic larvae compared to the axenic control (Figure 5C) in a magnitude similar to what we observed for Delfin 10X with CR larvae (see Figure 1). Taken together our data suggest that Delfin ingestion affects developmental time and growth independently of the commensal bacteria.

### 3.6. Btk Spores and Crystals Induce Midgut Perturbations

We previously showed that the ingestion of *Btk* vegetative cells by adult *Drosophila* reduced the accumulation of lipids in enterocytes [34], we wondered whether the commercial *Btk* product did the same in larvae. In unchallenged L3, lipid droplets strongly accumulated in enterocytes in the most anterior part of the midgut and weakly in the posterior midgut (Figure 6A) [68]. Instead of lipid droplet depletion, feeding the larvae with Delfin 10X provoked an accumulation of lipid droplets in the posterior larval midgut (Figure 6E). Furthermore, lipid droplets accumulated apically, facing the lumen (Figure 6F,F’ compared to Figure 6B,B’). Such an accumulation was not observed when developing larvae were fed with *Btk^ΔCry^* spores (Figure 6C,D,D’). Interestingly, it has been recently demonstrated that the pore-forming toxins hemolysin and monalysin produced by the enteropathogenic bacteria *Serratia marcescens* and *Pseudomonas entomophila* can induce an apical accumulation of lipid droplets in adult enterocytes of *Drosophila*, honeybees, and mice as well as in epithelial Caco-2 cells in culture. This phenomenon precedes the expulsion of cytoplasm required to get rid of toxins from the enterocytes [69]. Since the Cry toxins produced by *Btk* are also pore-forming toxins [8], we fed developing larvae with purified toxin crystals and checked for the accumulation of lipid droplets. Unexpectedly, we did not observe any accumulation of lipid droplets (Figure 6G,H,H’). However, feeding our developing larvae with naphthalene sulfonic acid (the additive present in commercial products, see Appendix A) provoked an accumulation of apical lipid droplets posterior to the acidic domain (Figure 6I,J,J’). To further assess whether such a lipid accumulation in the midgut would impair lipid metabolism in the whole larva, we quantified the circulating lipids (the diacylglycerides, DAG) in the hemolymph and the stored lipids (triacylglycerides, TAG) [44] in mid-third instar larvae. We did not notice any significant perturbation in lipid metabolism (Figure 6K) though a downward trend was visible for circulating DAG in larvae fed with Delfin 10X. Therefore, it is possible that such an apical accumulation of lipid droplets serves for the trap and elimination of naphthalene sulfonic acid similarly to what happens for the elimination of the *Sm* and *Pe* pore-forming toxins [69].

In adult *Drosophila*, pathogenic and opportunistic bacteria can induce enterocyte apoptosis upon ingestion [34,70]. Therefore, we verified the appearance of dying cells in the midgut using the Caspase3::GFP sensor to visualize apoptosis in living tissue [29] (see Section 2). While unchallenged *Btk^ΔCry^* fed larvae did not display any obvious apoptosis, the developing larvae fed with Delfin harbored apoptotic enterocytes mainly in the posterior part of the intestine (Figure 6L,M and Appendix A). We then checked whether the sealing of the gut epithelium was maintained despite the presence of apoptotic cells. We fed our developing larvae with the small fluorescent molecule Dextran^FITC^ that is unable to cross the midgut epithelium (Figure 6O) unless the intestinal barrier is broken, allowing the fluorescence to spread throughout the larva as in the case of *P. entomophila* infection (Figure 6P). Interestingly, *Btk^ΔCry^* spores as well as Delfin did not induce the loss of gut barrier function (Figure 6Q,R). Thus, our data show that *Btk* bioinsecticide hampers intestinal lipid homeostasis and induces enterocyte apoptosis without causing the rupture of the intestinal barrier. Such physiological perturbations most likely participate to the developmental delay and the growth defect observed.

### 3.7. Enterocyte Flattening and Incomplete Differentiation of Adult Midgut Precursors Help to Maintain Intestinal Integrity

Intestinal physiology can also be impeded by a disturbance of the apico-basal polarity of the gut epithelium. In the control L3 larval midgut, Armadillo (Arm, β-Catenin), a component of the adherens junction, marks the basolateral compartment of the enterocytes and Disc Large (Dlg), a component of the septate junction, marks the sub-apical compartment. Phalloidin that binds to F-Actin reveals both the apical brush border (facing the lumen) and the basal visceral muscles that surround the midgut lining (Figure 7A,A’). This organization is similar to the adult midgut [34,71]. Moreover, Arm also delineates the islets of the adult midgut precursors (AMPs), which will give rise to the adult midgut during metamorphosis (Figure 7A) [72,73,74]. When larvae were fed with the 10X concentration of *Btk^ΔCry^* spores, we did not observe any major perturbations of the architecture of late L3 midguts (Figure 7B,B’). However, we observed an increased number of AMPs per islet (Figure 7D) correlated with an increase in AMP division (Figure 7E). When larvae were fed with Delfin 10X, we also observed an increased number of AMPs by islet (Figure 7D) and a huge increase in AMP proliferation (Figure 7E) though at the expense of the number of islets (Figure 7F). Moreover, we also observed an increase in the enterocyte surface (Figure 7C,G). Interestingly, such an increase in enterocyte surface did not rely on a higher ploidy (Figure 7J) but rather on a change in cell shape, i.e., enterocyte flattening (Figure 7C’,H). Finally, we noticed the appearance of uncharacterized cells with an “intermediate” size. These cells that were intercalated between large enterocytes and the AMP islets, were neither mature polyploid enterocytes nor diploid AMPs or enteroendocrine cells (Figure 7I,I’). This observation was confirmed by analyzing the ploidy of the nuclei. In the control larvae, 89% of the polyploid cells underwent four endocycles of replication (32C nuclei, C corresponding to chromatin copy number) and 10% reached five cycles of endoreplication (64C) (Figure 7J). When larvae were raised on medium contaminated with *Btk^ΔCry^* spores, 74% corresponded to 32C nuclei and 25% to 64C nuclei (Figure 7J). When larvae were fed with Delfin 10X, a new population of cells with 16C, 8C and 4C nuclei appeared, representing 37% of the population of polyploid cells (Figure 7J). While the adult midgut quickly regenerates upon an oral infection thanks to the proliferation of adult intestinal stem cells and the differentiation of their daughter cells (the enteroblasts) into new enterocytes that repair the damages [75], such a mechanism has never been observed in *Drosophila* larvae. Indeed, the number of enterocytes composing the L3 intestine is defined during embryogenesis [76] and the larval gut grows without cell division but by increasing the ploidy and consequently the size of enterocytes. However, our data suggested that upon Delfin ingestion, new cells are formed. In the larval intestine, AMPs are the only cells capable of division [72,73,74]. Therefore, we hypothesized that the unknown cells with a nucleus ranging from 4C to 16C (Figure 7J) and adjacent to the AMP islets (Figure 7I) could arise from the differentiation of those AMPs. To test our hypothesis, we used the lineage tracing genetic tools developed in *Drosophila* called ReDDM [30]. The principle of this lineage tracing is based on the expression of both a labile GFP and a stable RFP within the AMPs thanks to the use of a promotor specifically expressed in AMPs and inducible at will (see Section 2). Accordingly, AMPs express both GFP and RFP, while the progenies, which are not AMPs, only express RFP. In control larvae, GFP and RFP were only present in the islet cells (Figure 7K). When larvae were fed with *Btk^ΔCry^* spores, GFP and RFP were also only expressed in the same AMP cells (Figure 7L). In contrast, when larvae were fed with Delfin 10X, cells harboring only RFP were now detectable. Furthermore, the newborn cells had nuclei of intermediate size ranging from 4C to 16C (Figure 7M). This result demonstrated that upon aggression by *Btk* bioinsecticides, the larval midgut is able to produce new cells, most probably immature enterocytes arising from AMPs. Altogether, our data suggest that the larval intestine maintains its integrity upon aggression through two mechanisms, (i) enterocyte flattening that increases the cell surface without new cycles of endoreplication and (ii) producing new immature enterocytes. Both mechanisms may allow the intestine to maintain the intestinal barrier, avoiding bacteria penetrating into the internal milieu and, therefore, preventing septicemia.

## 4. Discussion

In this study, we investigated the physiological and cellular mechanisms disturbed by *Btk* bioinsecticides, which are responsible for developmental delay and growth defects. Our data show that the impacts are mainly due to the synergy between *Btk* spores and crystals since either the spores (*Btk^ΔCry^*) or the purified crystals alone do not promote detectable developmental delay or growth defects in *Drosophila* larvae. This is in agreement with the study analyzing the safety of *Btk* purified Cry toxins on *Drosophila* larval development [18].

Interestingly, the effects we have characterized here are reminiscent of those observed on the target organisms *Helicoverpa armigera* and *H. zea*, *Spodoptera frugiperda* and *Choristoneura fumiferana*. Indeed, when fed with sublethal doses of *Btk* spores, all these target organisms display developmental delays and/or reduced pupal size [5,77,78,79]. Moreover, recent work has also shown that the potent pathogens *P. entomophila* and *P. aeruginosa* or huge amounts of the opportunistic bacterium *Erwinia carotovora carotovora* induce developmental delays and growth defects in *Drosophila* larvae [80]. These data suggest that developmental delays and/or growth defects must be general consequences of intestinal infections. The compromised gut digestive capacity of the infected larvae (whatever the insect) is likely the main reason for such developmental defects. Indeed, we have shown in larva (this work) and others in the adult [28,34,81,82,83] that allochthonous bacteria induce a more or less widespread apoptosis of enterocytes leading to decreased digestive and/or absorptive capacities. The developmental delay and the growth defect likely reflect the degree of damage suffered by the intestine. Houtz and colleagues observed a delay in pupation of up to four days using very virulent pathogens (Houtz et al., 2019) while we observed here a delay between 16 (Delfin) and 24 h (Dipel) with *Btk*. Therefore, while *Btk* behaves as a strong pathogen on target organisms, ultimately leading to death (in fact the expected effect), it has milder but significant impacts on the development of non-target insects, in accord with its opportunistic status.

Our work demonstrates that the composition of the diet is important to overcome *Btk* intestinal infection. Increasing the protein content of the nutritive medium rescues both growth defects and developmental delays. Our data are in agreement with previous observations in mammals and insects infected with various pathogenic bacteria or viruses showing that protein intake is important to withstand attacks by pathogens, to support an efficient immune defense and to promote growth, while intake of carbohydrates is deleterious for all these functions [57,58,59,84,85,86]. Moreover, our data show that the commensal bacterium *L. plantarum* also participates in the protection of larvae from *Btk* intestinal aggression in accordance with previous works involving commensal bacteria in gut protection against pathogens [87,88]. In *Drosophila*, *L. plantarum* has been shown to protect adult flies from *S. marcesens-* or *P. aeruginosa*-induced mortality and this protection is specific since the commensal bacteria *Enterococcus faecalis* or *B. subtilis* do not provide protection [89]. Our work strongly suggests that the benefit brought by *L plantarum* to the larvae enduring *Btk* aggression passes through an enhanced protein digestion and/or absorption by the gut in line with previous observations [28]. It would be interesting in the future to investigate the implication of both the TOR and the insulin receptor (InR) pathways in the phenotypes we observed. Indeed, the former acts in the fat body by sensing the amount free amino acids and remotely controls the secretion of Insulin Like Peptides (ILPs) by specialized cells located in the larval brain. The second is activated by circulating ILPs in peripheral tissues both to promote growth and to control duration of development (for review see [90,91] and references herein). Finally, we did not observe a decrease in food intake suggesting that spores and commercial products do not induce avoidance behavior. Together, our data support the model that the impact of the *Btk* products on the development of the non-target organism *D. melanogaster* (and probably many other arthropods) depends on both the nutritional environment and the performance of its commensal flora. Consequently, spraying *Btk* bioinsecticides when environmental conditions are stressful would amplify the adverse effects of *Btk*. Our work also sheds light on the pathogenic opportunism of *Btk* on a broad range of organisms owing to the conservation of the digestive tract architecture and functions [92,93]. For instance, the impacts of *Btk* on animals suffering from dietary deficiencies (very often associated with dysbiosis) may be more deleterious than on individuals with a well-balanced diet. Of note, bacteria of the *B. cereus* group are well-known opportunistic pathogens responsible for foodborne poisoning [45,94]. Therefore, a more accurate monitoring of the presence of *Btk* in food and environment is needed to avoid any setbacks, especially in the current context where organic farming is gaining market share.

Finally, our investigations have unraveled two concomitant mechanisms allowing *Drosophila* larvae to overcome an intestinal infection. It was thought so far that the *Drosophila* larval gut was unable to regenerate due to the short period covering the larval stages (about 5–6 days) and the inability to give rise to news cells to replenish the damaged ones [76]. Only two choices were thus offered to the larvae: *Live or let die*. However, we have uncovered a third choice for the infected larvae: *Die another day*. First, we demonstrate that the adult midgut precursors (AMPs) can be diverted from their original fate. Indeed, upon aggression of the larval intestinal epithelium, the AMPs can engage toward a process of differentiation to give rise to enterocyte-like cells, which are intercalated in between AMP islets and enterocytes. We can assume that such newborn cells could plug the holes caused by the death of enterocytes, thus avoiding the entry of luminal bacteria into the internal milieu. Unfortunately, newborn cells do not fully differentiate and probably do not participate in digestive functions as efficiently as well-differentiated enterocytes. The addition of these misdifferentiated cells to dying enterocytes can explain the reduced protease activity of the intestine of infected larvae and the subsequent developmental delay and growth defect. The increase in the proliferation of AMPs that we observed at the end of L3 upon *Btk* spore ingestion could be explained by the necessity to replenish the right number of AMPs before pupation to achieve a complete adult gut metamorphosis and, therefore, produce a viable adult fly. This AMP replenishment could also participate in the developmental delay we observed. The work recently published by Houtz and colleagues reaches similar conclusions [80].

We have also uncovered a second unexpected mechanism involved in the maintenance of gut integrity: The change in enterocyte shape. Previously, it was shown in many species (insects, mammals or plants) that the loss of cells in response to stressors can be compensated by the rapid increase in the size of neighboring cells thanks to successive endocycles [95]. Here, we evidence an increase in enterocyte area upon *Btk* spore ingestion but concomitantly we have observed a thinning of the epithelium. Above all, we did not observe an increase in the ploidy of the enterocyte nuclei. Interestingly, it has been shown that old enterocytes in adult *Drosophila* midgut cannot re-enter the endocycle [96]. Therefore, as for old adult enterocytes, L3 enterocytes (born during embryogenesis 7 days earlier) are probably unable to re-enter a cycle of endoreplication to promote cell growth. Hence, changing their shape through flattening increases the area covered by one enterocyte and participates in maintaining epithelium sealing. Here again, the absence of endoreplication likely participates in the reduced digestive capacities of the intestine and, therefore, in the developmental delays and growth defects.

It has been proposed that lepidopteran larvae succumb to *Btk* spores and toxins because they are unable to maintain their gut integrity. Conversely, for a lepidopteran larva to acquire resistance to *Btk* bioinsecticides may rely on its capacity to maintain/regenerate its intestinal integrity [97,98,99]. Non-target organisms do not die likely because they are able to regenerate their gut lining faster than *Btk* bioinsecticides destroy it. However, increasing the amount of *Btk* provided to *Drosophila* larvae ultimately leads to their death [22,100], suggesting that the harmfulness of *Btk* bioinsecticides relies on both the amounts of *Btk* ingested and the affinity of Cry toxins for more or less specific receptors. In agreement, the Cry2A toxins present in *Btk* bioinsecticides display cross-order toxicity toward larvae of certain mosquito species [9] (and references herein).

## 5. Conclusions

Our work highlight the putative unintended impacts that an increase use of *Btk* bioinsecticides may have on non-target insects. The opportunistic virulence of *Btk* for many organisms should not be overlooked, especially for those suffering from nutritional stress, dysbiosis or with a compromised innate immunity. A situation that more and more species will encounter with the climatic changes occurring worldwide. Many investigations remain to be done to improve the use and efficiency of *Btk* spores and toxins and to make their use as safe as possible for both handlers, consumers, and the environment.

## Figures and Tables

**Figure 1 insects-11-00697-f001:**
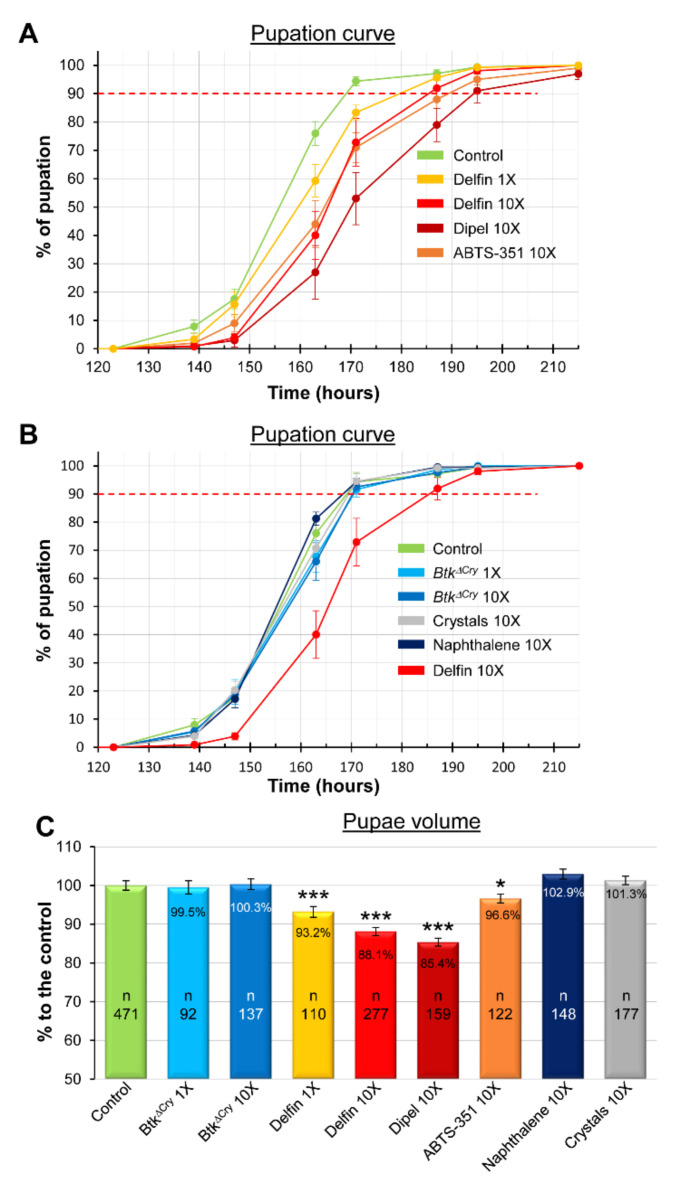
*Btk* bioinsecticide impacts on developmental delay and growth defects. (**A**) Pupation curve of larvae fed with commercial products (Delfin and Dipel) or with a homemade preparation of spores of the *Btk* strain ABTS-351 (Dipel). 152 < *n* < 262. In this figure and in all the following figures, the cumulative percentage of pupation is shown over time. (**B**) Pupation curve of larvae fed with a *Btk* strain devoid of Cry toxins (*Btk^ΔCry^*) or with Cry toxin containing crystals purified from Dipel or with naphthalene sulfonic acid at a concentration equivalent to the amount found in Delfin 10X. In (**A**,**B**), red dashed lines mark 90% of the population reaching pupation. 143 < *n* < 263. (**C**) Pupae volume in the different conditions listed on the x-axis. *n* = number of individuals counted. For the controls, n corresponds to the total number of pupae measured when one compiled all the internal controls. Error bars represent the standard error of the mean (SEM). * *p* < 0.05; *** *p* < 0.001 compared with controls.

**Figure 2 insects-11-00697-f002:**
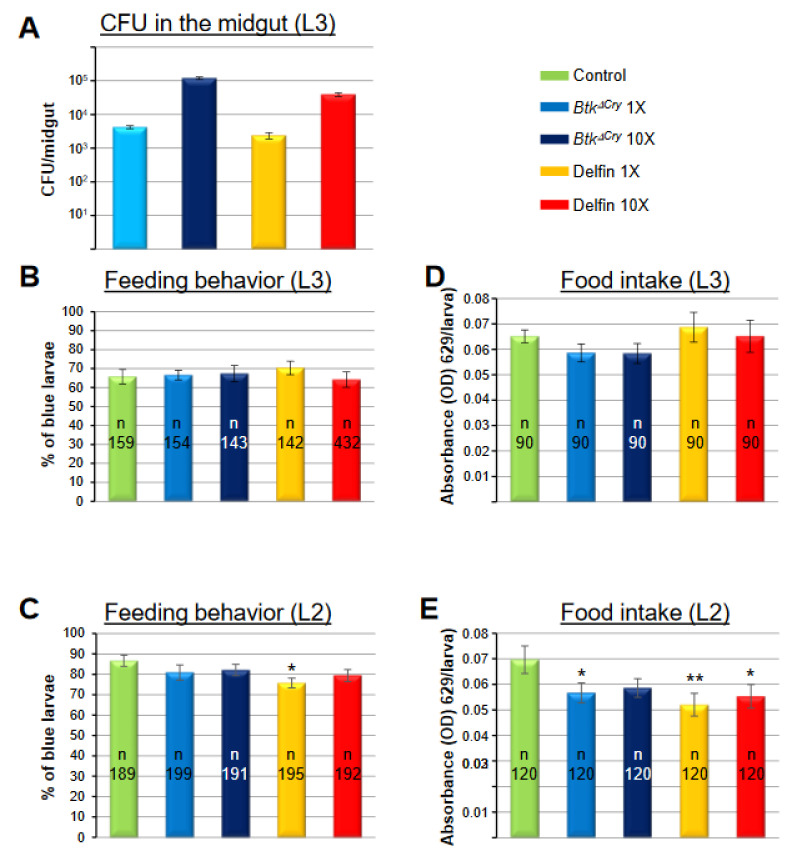
*Btk* bioinsecticides and larval feeding behavior. (**A**) L3 midgut load of larvae fed with either spores of *Btk^ΔCry^* or Delfin. (**B**,**C**) Percentage of L3 (**B**) and L2 (**C**) presenting blue dyed food in the intestine. (**D**,**E**) Amount of dye ingested by L3 (**D**) and L2 (**E**). Green bars: Control larvae (fed with water); light blue bars: 1X concentration of *Btk^ΔCry^* spores; dark blue bars: 10X concentration of *Btk^ΔCry^* spores; yellow bars: 1X concentration of Delfin; Red bars: 10X concentration of Delfin. *N* = number of individuals analyzed. Errors bars represent the SEM. * *p* < 0.05, ** *p* < 0.01 compared with controls.

**Figure 3 insects-11-00697-f003:**
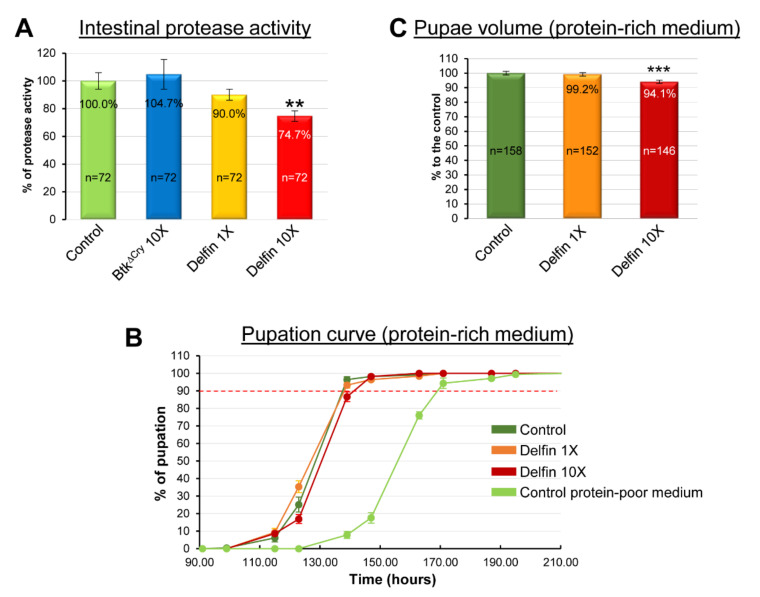
Influence of a protein-rich-diet on Delfin impacts (**A**) Measurement of the protease activity of the intestine of L3 raised on control medium contaminated with H_2_O (control) or with 10X concentrations of spores of *Btk^ΔCry^* or Delfin or 1X concentration of spores of Delfin. (**B**) Pupation curve of larvae raised on a protein-rich medium (10% yeast instead of 2%) contaminated with 1X or 10X concentrations of Delfin. Note the shortening of the developmental time of the control larvae raised on the protein-rich medium (10% yeast) compared to larvae raised on the protein-poor-medium (2% yeast). (**C**) Pupae volume derived from larvae raised on the protein-rich medium and in the different conditions listed on the x-axis. Errors bars represent the SEM. ** *p* < 0.01, *** *p* < 0.001 compared with controls.

**Figure 4 insects-11-00697-f004:**
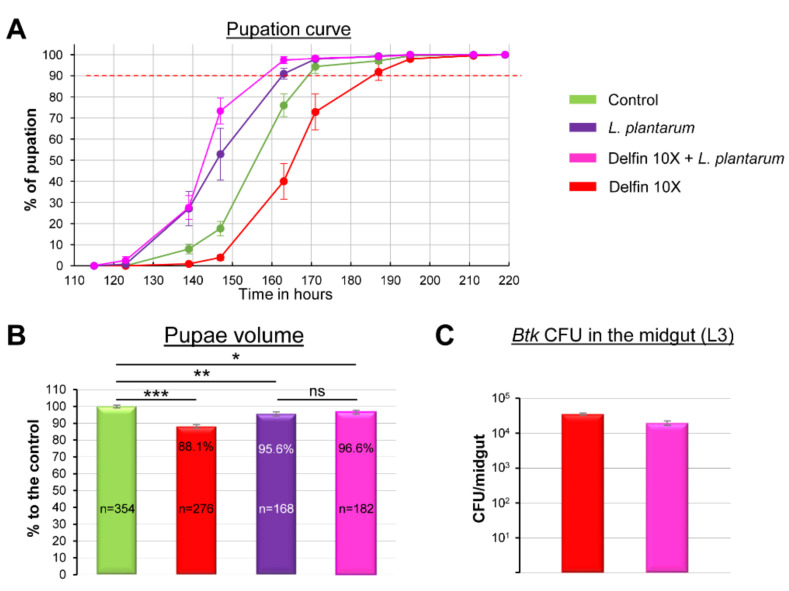
*L. plantarum activity* helps to overcome *Btk* bioinsecticide impacts (**A**) Pupation curve of larvae reared on a medium complemented with 10^6^ CFU of *L. plantarum* and contaminated with Delfin 10X (pink) or not (purple) compared to control larvae (green) or Delfin 10X only (red). 144 < *n* < 214. (**B**) Pupae volume derived from larvae raised on a medium complemented with 10^6^ CFU of *L plantarum* and contaminated with Delfin 10X (pink) or not (purple) compared to control pupae reared on control medium (green) or medium contaminated with Delfin 10X only (red). (**C**) *Btk* load in the midgut of L3 reared on a medium contaminated with Delfin 10X and complemented (pink) or not (red) with 10^6^ CFU of *L. plantarum*. Errors bars represent the SEM. ns means not significative, * *p* < 0.05, ** *p* < 0.01, *** *p* < 0.001 compared with controls.

**Figure 5 insects-11-00697-f005:**
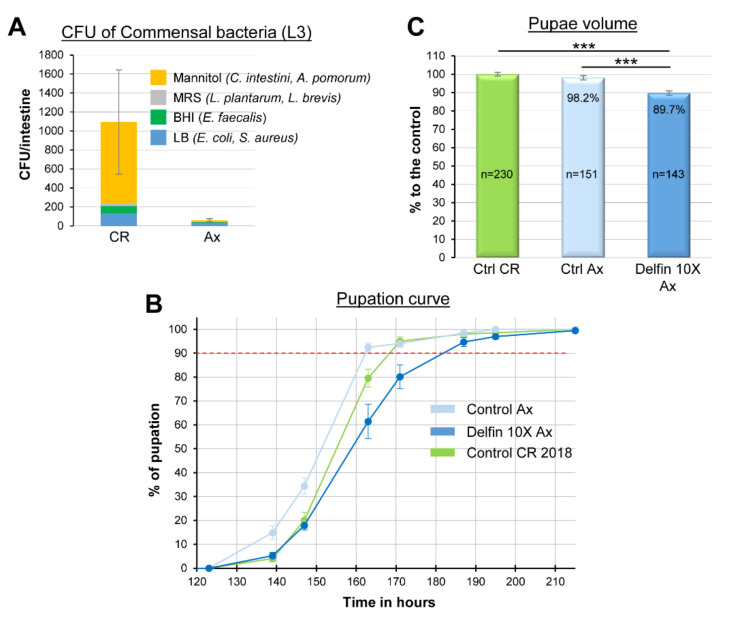
*Btk* bioinsecticides interaction with commensal flora (**A**) Commensal bacterial load in the midgut of L3 of Drosophila Canton S strain reared on conventional medium (CR, left bar) or reared in axenic condition (Ax, right bar). The *Commensalibacter intestini* and *Acetobacter pomorum* grow on mannitol plate (orange), *L plantarum* and *L brevis* grow on MRS plate (gray), *Enterococcus faecalis* grows on BHI plate (green) and *Escherichia coli* and *Staphylococcus aureus* grow on LB plate (blue). (**B**) Pupation curve of axenic larvae raised on a medium contaminated with Delfin 10X (dark blue) or not (light blue, control Ax) compared to larvae conventionally reared (CR, green). 171 < *n* < 211. (**C**) Pupae volume derived from conventionally reared larvae (green) or from axenic larvae raised on a sterile medium contaminated with Delfin 10X (dark blue) or not (light blue). Errors bars represent the SEM. *** *p* < 0.001 compared with controls.

**Figure 6 insects-11-00697-f006:**
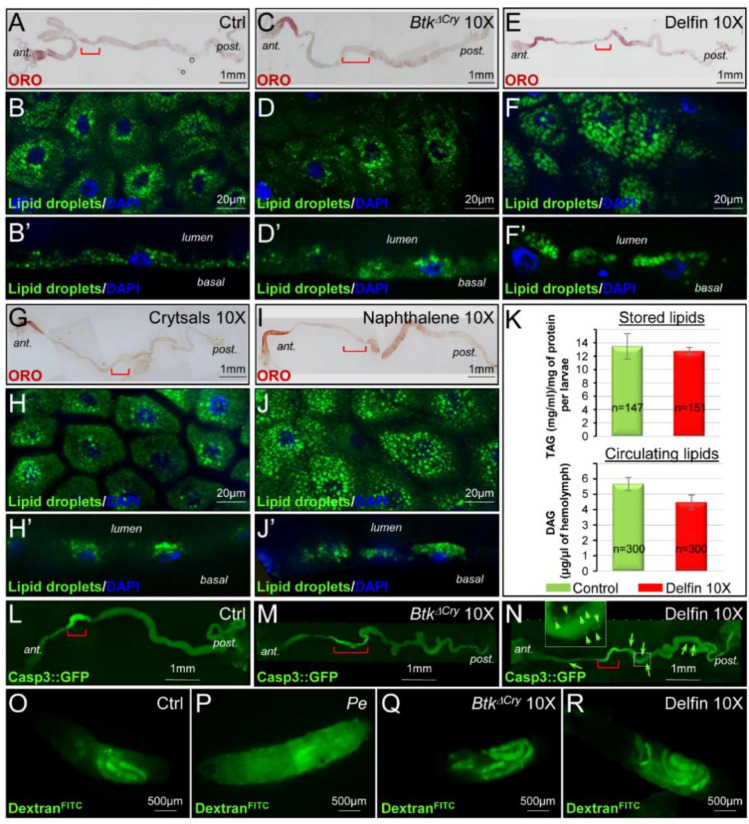
*Btk* bioinsecticides impacts on gut integrity. L3 raised on a control medium (**A**–**B’**) or a medium contaminated with *Btk^ΔCry^* 10X (**C**–**D’**), Delfin 10X (**E**,**F**), purified Crystals 10X (**G**–**H’**), naphthalene sulfonic acid 10X (**I**–**J’**). (**A**–**J’**) Labeling of lipid droplets with ORO (red, (**A**,**C**,**E**,**G**,**I**)) or with Bodipy (green, (**B**,**B’**,**D**,**D’**,**F**,**F’**,**H**,**H’**,**J**,**J’**)). DAPI (4′,6-diamidino-2-phenylindole) staining in blue marks the nuclei. Midguts are oriented anterior to the left. Red brackets delineate the acidic region of the midgut. (**B’**,**D’**,**F’**,**H’**,**J’**) are x/z section with the apical pole of the epithelium facing the lumen. (**K**) Stored (TAG, upper panel) and circulating (DAG, lower panel) lipid dosages in control larvae (green bars) and in larvae raised on Delfin 10X contaminated medium (red bars). (**L**–**N**) myo1A > Casp::GFP larvae raised on control medium (**L**) or on a medium contaminated with *Btk^ΔCry^* 10X (**M**) or Delfin 10X (**N**). Arrows point to enterocytes in which Caspase 3 is activated. (**O**–**P**) L3 raised on a medium complemented with Dextran^FITC^ (**O**) or with Dextran^FITC^ and contaminated with *P. entomophila* (*Pe*) (**P**), with *Btk*^Δ*Cry*^ 10X (**Q**) or Delfin 10X (**R**).

**Figure 7 insects-11-00697-f007:**
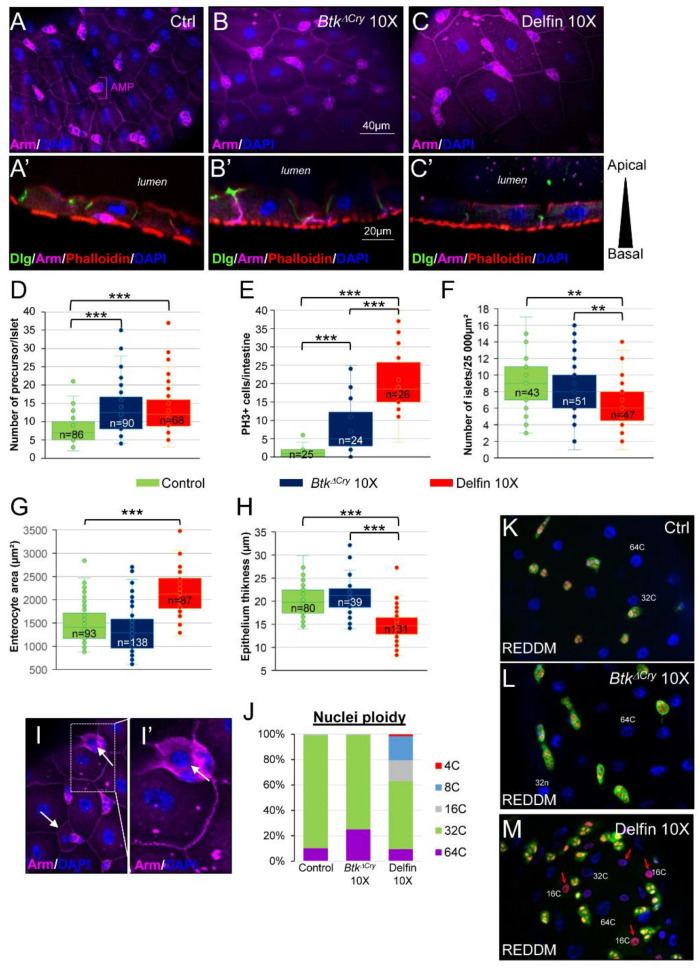
Larval midgut regeneration upon aggression. (**A**–**C’**) Immunolabelling of Dlg::GFP L3 midgut from larvae raised on control medium (**A**,**A’**), on medium contaminated with *Btk^ΔCry^* 10X (**B**,**B’**) or Delfin 10X (**C**,**C’**). Midguts were stained with the adherens junction marker Armadillo (Arm, purple) that strongly delineates the adult midgut precursors (AMP) (**A**,**B**,**C**) and marks the basolateral compartment of the enterocytes (**A’**,**B’**,**C’**). DAPI (blue) marks the nuclei and Phalloidin (red) labels the visceral muscle surrounding the intestine and the apical pole of the enterocytes facing the lumen (in (**A’**,**B’**,**C’**)). Dlg::GFP (green) marks the sub apical compartment of the enterocytes. (**D**–**H**) L3 midgut from larvae raised on control medium (green bars), on medium contaminated with *Btk^ΔCry^* 10X (dark blue bars) or Delfin 10X (red bars). (**D**) Number of AMPs per islet, (**E**) number of dividing AMPs per intestine (mitotic AMPs are labeled by the Phospho-Histone 3 (PH3) immunostaining), (**F**) Number of AMP islets (an islet contains more than one diploid cell) per unit of surface, (**G**) enterocyte area and (**H**) epithelium thickness in the posterior midgut (3 independent measures were taken by image capture). ((**I**) and enlargement in (**I’**)) Dlg::GFP L3 midgut from larvae raised on medium contaminated with Delfin 10X and stained for Arm (purple) and DAPI (blue). Arrows point to the cells with intermediate size (smaller than enterocytes but bigger than AMPs). (**J**) Assessment of enterocyte ploidy in posterior L3 midgut from larvae raised on control medium, on medium contaminated with *Btk^ΔCry^* 10X or Delfin 10X (C corresponding to chromatin copy number). (**K**–**M**) Lineage tracing of AMP progeny in the w; esg-Gal4, UAS-GFP/+; UAS-H2B::RFP, tub-Gal80^ts^/+ (esg ReDDM) L3 midgut from larvae raised on control medium (**K**), on medium contaminated with *Btk*^Δ*Cry*^ 10X (**L**) or Delfin 10X (**M**). DAPI (blue) marks the nuclei. ** *p* < 0.01, *** *p* < 0.001 compared with controls.

**Table 1 insects-11-00697-t001:** Time in hours when 10%, 50%, and 90% larvae enter the pupation stage (immobilized) expressed in hours after egg laying. “n” corresponds to the number of larvae counted in each condition. Please note that we compiled the controls per year of experiments (encompassing years from 2017 to 2019).

% of Pupation	10%	50%	90%	*n*
Control 2017	140.5	156.0	169.0	183
Control 2018	142.0	155.0	168.5	211
Control 2019	142.5	155.5	171.0	203
Delfin 1X	143.0	159.5	179.5	262
Delfin 10X	149.5	165.5	185.0	214
Dipel 10X	151.5	170.0	194.0	152
ABTS-351 10X	147.5	165.0	189.0	159
*Btk^ΔCry^* 1X	141.5	157.0	170.5	263
*Btk^ΔCry^* 10X	141.5	157.5	170.5	204
Naphthalene sulfonic acid	142.5	155.0	168.0	212
Crystals	142.0	156.5	169.5	143
Control protein-rich medium	117.0	128.5	137.5	212
Delfin 1X protein-rich medium	115.0	127.0	138.0	195
Delfin 10X protein-rich medium	117.0	130.5	141.0	207
*L. plantarum*	128.0	146.0	162.5	144
Delfin 10X + *L plantarum*	127.0	143.0	158.0	153
Control Axenic	134.0	151.5	162.5	171
Delfin 10X Axenic	142.0	159.0	182.0	172

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
