# Peer review of "Bacillus thuringiensis Bioinsecticides Induce Developmental Defects in Non-Target Drosophila melanogaster Larvae"

_insects, 2020, doi:10.3390/insects11100697_

Round 1

Reviewer 1 Report

This manuscript by Nawrot-Esposito and colleagues examines the effects of the bioinsecticidal bacteria Bacillus thuringiensis on Drosophila melanogaster larval development and intestinal integrity. Through a series of convincing experiments, the authors demonstrate that a common insecticide (Delfin) that contains this bacteria disrupts D. melanogaster larval development – and interesting result considering that flies are thought to be resistant to such treatments. Moreover, the authors demonstrate that bacteria unable to produce cry toxin, as well as isolated cry toxin, do not induce a developmental delay, suggesting that the combination of bacteria + toxin are important for the growth delay. Intriguingly, Delfin toxicity can be overcome by increasing the amount of protein within the diet, hinting at a potential mechanism that drives the off-target toxic affects of this insecticide. Finally, the authors then demonstrate that that Delfin affects intestinal physiology and morphology.

Overall, I found this an interesting study that raises important questions about this class of insecticide. The experiments seem well controlled and the conclusions important. I could see this manuscript being highly cited. I have few comments and believe the study requires no additional experiments. I request that the authors address my minor concerns:

Line 209: “sugar content delays development…” While this is true when excessive sugar is added to the diet, I’d note that sugar is necessary for development. The authors should use a more nuanced statement.

Lines 231-237: The observation that increased dietary protein can overcome Delfin treatment is quite interesting. Since larval insulin signaling is triggered by nutritional amino acid content, and decreased insulin signaling delays larval development, the authors should speculate about the potential link between Delfin and insulin in the discussion. I’d also encourage the authors to examine this possibility in future studies.

Line 295: remove the word “strongly” before “suggest.” The data stands on its own. I could easily suggest alternative models and using “strongly” makes me wonder why these are not considered.

Line 349: “Such physiological most…” Seems like the authors are missing a point.

Line 484: “Fisrt” is spelled incorrectly.

Line 492: Please remove the last word of this line “huge” This is an unnecessary adjective that adds no value – how do you define huge in terms of significance.

Author Response

Comments and Suggestions for Authors

This manuscript by Nawrot-Esposito and colleagues examines the effects of the bioinsecticidal bacteria Bacillus thuringiensis on Drosophila melanogaster larval development and intestinal integrity. Through a series of convincing experiments, the authors demonstrate that a common insecticide (Delfin) that contains this bacteria disrupts D. melanogaster larval development – and interesting result considering that flies are thought to be resistant to such treatments. Moreover, the authors demonstrate that bacteria unable to produce cry toxin, as well as isolated cry toxin, do not induce a developmental delay, suggesting that the combination of bacteria + toxin are important for the growth delay. Intriguingly, Delfin toxicity can be overcome by increasing the amount of protein within the diet, hinting at a potential mechanism that drives the off-target toxic affects of this insecticide. Finally, the authors then demonstrate that that Delfin affects intestinal physiology and morphology.

Overall, I found this an interesting study that raises important questions about this class of insecticide. The experiments seem well controlled and the conclusions important. I could see this manuscript being highly cited. I have few comments and believe the study requires no additional experiments. I request that the authors address my minor concerns:

Line 209: “sugar content delays development…” While this is true when excessive sugar is added to the diet, I’d note that sugar is necessary for development. The authors should use a more nuanced statement.

We have changed the sentence. We have now written "Proteins are a key nutrient to drive normal larval development and growth. Increasing protein content in the diet accelerates the developmental time and/or stimulates growth, while an excess of sugar delays development and a change in lipid amounts has only a marginal impact"

Lines 231-237: The observation that increased dietary protein can overcome Delfin treatment is quite interesting. Since larval insulin signaling is triggered by nutritional amino acid content, and decreased insulin signaling delays larval development, the authors should speculate about the potential link between Delfin and insulin in the discussion. I’d also encourage the authors to examine this possibility in future studies.

We have added a small part Lines 472-478: "It would be interesting in the future to investigate the implication of both the TOR and the insulin receptor (InR) pathways in the phenotypes we observed. Indeed, the former acts in the fat body by sensing the amount free amino-acids and remotely controls the secretion of Insulin Like Peptides (ILPs) by specialized cells located in the larval brain. The second is activated by circulating ILPs in peripheral tissues both to promote growth and to control duration of development (for review see (Koyama et al., 2020; Manière et al., 2020) and references herein)."

Line 295: remove the word “strongly” before “suggest.” The data stands on its own. I could easily suggest alternative models and using “strongly” makes me wonder why these are not considered.

Done

Line 349: “Such physiological most…” Seems like the authors are missing a point.

Ooops soory. we have now corrected the sentence (line 356)

"Such physiological perturbations most likely perturbations participate to the developmental delay and the growth defect observed."

Line 484: “Fisrt” is spelled incorrectly.

Corrected

Line 492: Please remove the last word of this line “huge” This is an unnecessary adjective that adds no value – how do you define huge in terms of significance.

Removed

Reviewer 2 Report

I have made many suggestions in the attached document to improve the format and consistency of your manuscript. I can see that an extreme amount of work has been done and a huge number of experiments needed to be described. However, there are many inconsistencies throughout that I have noted.

A few bigger/repeated ones:

  1. Is it acceptable with the journal to have methods last, if not the methods and results will need extensive revision due to the change of order
  2. The references need to be converted from author date to numerical
  3.  You refer to dose, which is only appropriate if you know how much entered your larvae, otherwise it is concentration, should be changed throughout
  4. Make sure your figure legends have the insect species name and any abbreviations defined
  5. You refer to L3 and larvae in the same sentence but i think they are one and the same as L3 stands for 3rd instar larvae? Needs revision throughout

Author Response

Comments and Suggestions for Authors

I have made many suggestions in the attached document to improve the format and consistency of your manuscript. I can see that an extreme amount of work has been done and a huge number of experiments needed to be described. However, there are many inconsistencies throughout that I have noted.

We are grateful to the reviewer for all these corrections. We hope that most of them have been fixed. Nevertheless, we observed that of the typographic errors came from the word to PDF conversion on the website. Indeed, these errors are absent from our original Word (.docx) manuscript.

A few bigger/repeated ones:

1. Is it acceptable with the journal to have methods last, if not the methods and results will need extensive revision due to the change of order

I do not know. The formatted file of our manuscript has been provided by the editorial office. So, if it is necessary to put Methods before the results, we will do the change once the manuscript will be definitively accepted. Indeed, for the answers to the reviewers, we used the line numbers of the editorial office's format that the reviewers received. This facilitates the follow-up of the corrections we have done.

2 .The references need to be converted from author date to numerical

As above. We will wait for the final version before changing the references. We have down loaded the Endnote MDPI format.

3. You refer to dose, which is only appropriate if you know how much entered your larvae, otherwise it is concentration, should be changed throughout

The reviewer is right. We have changed the word in the manuscript.

Noteworthy, we have estimated the amount of Bt ingested by our larvae in figure 2A and in figure 4C.

4. Make sure your figure legends have the insect species name and any abbreviations defined

We have corrected many small typing mistakes in the figure legends. Some of them like BtkΔCry which was not correctly written, probably arose from the conversion of our manuscript on the website. Indeed, those typing mistakes were not present in our original word file.

5. You refer to L3 and larvae in the same sentence but i think they are one and the same as L3 stands for 3rd instar larvae? Needs revision throughout

Right. We have now defined L3 line 186: "When we estimated the amount of Btk cells in the midgut of third instar larvae (L3) (Fig. 2A), no….". We have also corrected this redundancy all along the text.

Reviewer 3 Report

The manuscript describes the effects of Bacillus thuringiensis subsp. kurstaki (Btk) on the development and growth of Drosophila melanogaster larvae. Authors conclude that Btk induces developmental and growth defects due to the disturbance of the larval intestinal physiology. 

This reviewer wonders why authors pick D. melanogaster for their non-target study. As they state in Introduction, it is a well-established model organism. However, it is a common pest in homes, restaurants and other places where food is served.

In addition, one of the Btk strains such as HD-1, the active ingredient of commercial insecticide controlling mainly lepidopteran crop pests, produces Cry2Aa along with Cry1Aa, Cry1Ab and Cry1Ac, and forms two different crystals during sporulation. Unlike Cry1A, Cry2Aa is known to have dual toxicity against lepidopteran and dipteran larvae although the toxicity against dipteran larvae is not high. One of the citations in this manuscript (Cossentine et al. 2016) demonstrated this. Therefore, the results presented in the manuscript does not surprise this reviewer.

More importantly, although it is nice to learn, the results obtained in this manuscript cannot reflect what really happens in the field as there are many environmental factors affecting Btk's activity and other non-target insects in the field. This reviewer suggests authors tone down the conclusion significantly. 

Line 28: Replace "var." with "subsp." No one uses "var." any more. Check the manuscript thoroughly for this.

Lines 33-34: What do authors mean by saying "Btk bacteria and Btk insecticidal toxins"? Would it be spores and toxins?

Figure 1 A and B, and part of Table 1 overlap the data presented. Delete Figure 1 A and B (or corresponding part of Table 1). The rows starting "Control protein-rich medium" to all the way down in Table 1 do not belong to this part of the results. Make a separate Table for this part for the section 2.5.

For the similar reason, separate Figure 4 A, B and C, and D, E and F. The former belongs to the section 2.4 and the latter to the section 2.5.

Figure 6: Authors need to include another bacterial species such as Pseudomonas entomophila as a control strain like in Figure 5 in order to support lines 347-349.

Author Response

Comments and Suggestions for Authors

The manuscript describes the effects of Bacillus thuringiensis subsp. kurstaki (Btk) on the development and growth of Drosophila melanogaster larvae. Authors conclude that Btk induces developmental and growth defects due to the disturbance of the larval intestinal physiology. 

This reviewer wonders why authors pick D. melanogaster for their non-target study. As they state in Introduction, it is a well-established model organism. However, it is a common pest in homes, restaurants and other places where food is served.

The reviewer is right, D. melanogaster can be considered as a pest. However, in this study we are interested in the physiological and cellular mechanism affected by the ingestion of Btk bioinsecticides. Btk does not primarily target Drosophila larvae. To kill D. melanogaster larvae we have to increase the dose by 100 (109 CFU/g) (Babin et al., 2020). Here we use D. melanogaster for the availability of the cellular tools. It is for this reason we used the word "model organism". In our manuscript, we also consider D. melanogaster as a non-target insect. So, using the word pest would generate confusion for the reader regarding the lepidopteran that are Btk targets.

Babin, A., Nawrot-Esposito, M.-P., Gallet, A., Gatti, J.-L. and Poirié, M. (2020). Differential side-effects of Bacillus thuringiensis bioinsecticide on non-target Drosophila flies. Sci Rep 10, 16241.

In addition, one of the Btk strains such as HD-1, the active ingredient of commercial insecticide controlling mainly lepidopteran crop pests, produces Cry2Aa along with Cry1Aa, Cry1Ab and Cry1Ac, and forms two different crystals during sporulation. Unlike Cry1A, Cry2Aa is known to have dual toxicity against lepidopteran and dipteran larvae although the toxicity against dipteran larvae is not high. One of the citations in this manuscript (Cossentine et al. 2016) demonstrated this. Therefore, the results presented in the manuscript does not surprise this reviewer.

The reviewer is right. The strain SA-11 (Delfin) and ABTS-351 (Dipel) produce Cry2Aa and Cry2Ab. However as stated just above, to get larval lethality we have to increase the dose to reach 109 CFU/g. Even if the data presented in this manuscript are not a surprise for the reviewer, this has never been demonstrated before. Moreover, we also decipher the physiological processes and the cellular mechanisms allowing the larvae to overcome Btk ingestion. This also has never been demonstrated. All the results presented in this work are new.

Finally, it is admitted that Btk specifically kills lepidopteran larvae at the doses recommended by manufacturers. Here we show that, in fact, at these doses Btk bioinsecticides may affect other orders than Lepidoptera, even if there is no lethality.

We have added a sentence in the introduction (lines 73-74) and in the discussion (line 532-533) regarding this subject.

More importantly, although it is nice to learn, the results obtained in this manuscript cannot reflect what really happens in the field as there are many environmental factors affecting Btk's activity and other non-target insects in the field. This reviewer suggests authors tone down the conclusion significantly. 

We do not claim that our work reflects what is happening in the field. Lines 97-100 we write "Here, using two realistic concentrations, a concentration detected on vegetables after one spraying and a concentration only tenfold higher (but equivalent to the ten cumulative treatments authorized by the European Union), we observed that Btk bioinsecticides induced developmental delay and reduced growth of D. melanogaster larvae." We are interested in the cellular and physiological mechanisms. To understand what is happening, we used doses that are recovered on vegetables after spraying. We think that this approach is more physiologic than the use of huge doses that will strongly disturb the intestine and would likely mask the physiological effects (and even would kill the larvae).

In all the small conclusions at the end of each part of the Results section, we always conclude on a mechanistic way, never as an environmental way. We do the same in the abstract and at the end of the introduction.

Line 28: Replace "var." with "subsp." No one uses "var." any more. Check the manuscript thoroughly for this.

We did it.

Lines 33-34: What do authors mean by saying "Btk bacteria and Btk insecticidal toxins"? Would it be spores and toxins?

Bacteria can be either vegetative cells or spores. Bacteria is a generic term. As we do not know yet whether the effects are due spores, vegetative cells or both. We, therefore, use the word bacteria that encompasses both forms.

Figure 1 A and B, and part of Table 1 overlap the data presented. Delete Figure 1 A and B (or corresponding part of Table 1). The rows starting "Control protein-rich medium" to all the way down in Table 1 do not belong to this part of the results. Make a separate Table for this part for the section 2.5.

We do this voluntary. Table 1 recapitulates all the conditions we have studied. It is easier for the reader to compare the different conditions and to make his/her own interpretation. Moreover we also add the analyses for 10% and 50% of pupation that are absent from the figure 1. So, we would like to keep all the data in a single table to facilitate comparisons between conditions. Moreover, the conditions are listed, in the Table 1, in the order they are appearing in the text.

For the similar reason, separate Figure 4 A, B and C, and D, E and F. The former belongs to the section 2.4 and the latter to the section 2.5.

We have separates the figure. It is now figure 4 and figure 5.

Figure 6: Authors need to include another bacterial species such as Pseudomonas entomophila as a control strain like in Figure 5 in order to support lines 347-349.

P.entomophila ultimately kills the larvae. It is impossible to dissect the gut of dying larvae. Too fragile.

Lines 354-356 we write "Thus, our data show that Btk bioinsecticide hampers intestinal lipid homeostasis and induces enterocyte apoptosis without causing the rupture of the intestinal barrier". We only conclude for Btk bioinsecticides, not for P entomophila.

Moreover lines 354-357 are the conclusion for old figure 5 (now figure 6 in the new version). So, I do not understand the link between the conclusion of old figure 5 and old figure 6 (now figure 7).

Round 2

Reviewer 3 Report

I think all of my comments are addressed.